# Fecal Immunochemical Tests Detect Screening Participants with Multiple Advanced Adenomas Better than T1 Colorectal Cancers

**DOI:** 10.3390/cancers13040644

**Published:** 2021-02-05

**Authors:** Anton Gies, Tobias Niedermaier, Laura Fiona Gruner, Thomas Heisser, Petra Schrotz-King, Hermann Brenner

**Affiliations:** 1Division of Preventive Oncology, German Cancer Research Center (DKFZ) and National Center for Tumor Diseases (NCT), Im Neuenheimer Feld 460, D-69120 Heidelberg, Germany; petra.schrotz-king@nct-heidelberg.de (P.S.-K.); h.brenner@dkfz-heidelberg.de (H.B.); 2Division of Clinical Epidemiology and Aging Research, German Cancer Research Center (DKFZ), D-69120 Heidelberg, Germany; t.niedermaier@dkfz-heidelberg.de (T.N.); l.gruner@dkfz-heidelberg.de (L.F.G.); t.heisser@dkfz-heidelberg.de (T.H.); 3Medical Faculty Heidelberg, University of Heidelberg, D-69120 Heidelberg, Germany; 4German Cancer Consortium (DKTK), German Cancer Research Center (DKFZ), D-69120 Heidelberg, Germany

**Keywords:** colon cancer, fecal occult blood test, advanced neoplasia, early detection, prevention

## Abstract

**Simple Summary:**

Fecal occult blood tests (FOBTs) detect colorectal cancer (CRC) at high levels of sensitivity and specificity. However, the detection of early-stage cancers is highly important to reduce CRC mortality. We aimed to assess the sensitivity of a large number of different FOBTs according to various tumor characteristics. We observed among all FOBTs consistently lower sensitivities for UICC stage I cancers in comparison to more advanced cancer stages. An even stronger gradient was found according to T status, with substantially lower sensitivities for T1 than for T2–T4 cancers. Furthermore, sensitivities for T1 cancers were even lower than sensitivities for detection of multiple advanced adenomas. Further research should focus on improving the sensitivity of non-invasive tests for detection of UICC stage I and T1 cancers.

**Abstract:**

Background: Fecal immunochemical tests (FITs) are widely used for colorectal cancer (CRC) screening. The detection of early-stage cancer and advanced adenoma (AA), the most important premalignant lesion, is highly relevant to reducing CRC-related deaths. We aimed to assess sensitivity for the detection of CRC and AA stratified by tumor stage; number; size; histology of AA; and by location, age, sex, and body mass index (BMI). Methods: Participants of screening colonoscopy (*n* = 2043) and newly diagnosed CRC patients (*n* = 184) provided a stool sample before bowel preparation or CRC surgery. Fecal hemoglobin concentration was determined in parallel by nine different quantitative FITs among 94 CRC patients, 200 AA cases, and 300 participants free of advanced neoplasm. Sensitivities were calculated at original cutoffs and at adjusted cutoffs, yielding 93% specificity among all FITs. Results: At adjusted cutoffs, UICC stage I cancers yielded consistently lower sensitivities (range: 62–68%) compared to stage II–IV cancers (range: 73–89%). An even stronger gradient was observed according to T status, with substantially lower sensitivities for T1 (range: 39–57%) than for T2–T4 cancers (range: 71–100%). Sensitivities for the detection of participants with multiple AAs ranged from 55% to 64% and were by up to 25% points higher than sensitivities for T1 cancers. Conclusions: FITs detect stage I cancers and especially T1 cancers at substantially lower sensitivities than more advanced cancer stages. Participants with multiple AAs were detected with slightly lower sensitivities than stage I cancers and with even higher sensitivities than T1 cancers. Further research should focus on improving the detection of early-stage cancers.

## 1. Introduction

Fecal immunochemical tests (FITs) for hemoglobin are widely used for colorectal cancer (CRC) screening [1,2,3]. FITs achieve overall high sensitivities for the detection of CRC, in the range of 70–80% at very high specificities of 90–95% [4,5,6]. Detection of advanced adenoma (AA), the most important precursor, and early-stage cancers is highly relevant for the reduction of CRC mortality [7,8]. However little is known with respect to the detection of CRC and AA stratified by various characteristics.

Two recent publications [9,10] reported that sensitivities for CRC detection tended to be higher with more advanced UICC stage and differences were suggested to be particularly strong according to T status. However, previous comparisons of stage-specific FIT-performance were all performed using one FIT brand each, and specificities as well as sensitivities for the detection of subgroup-specific CRCs and AAs varied widely between the different studies [9]. It is not clear to what extent these variations reflect differences between the various FITs or relate to differences in populations and designs of the studies. 

In this study, we simultaneously assessed the sensitivity of nine different FIT brands according to tumor stage (UICC stage or T status) and location; size; histology; and number of AAs; and sex, age, and body mass index (BMI) among participants of screening colonoscopy and newly diagnosed CRC cases.

## 2. Methods

We followed the Standards for the Reporting of Diagnostic Accuracy Studies (STARD) [11] and the guideline for Faecal Immunochemical Tests for Haemoglobin Evaluation Reporting (FITTER) [12].

### 2.1. Study Design and Population

This study is based on the BLITZ and DACHSplus studies. Detailed information about both studies has been provided elsewhere [13,14,15,16]. Briefly, the BLITZ study is an ongoing prospective study among participants of screening colonoscopy, who are recruited before their scheduled colonoscopy appointment. The recruitment of study participants was conducted by 20 cooperating study sites and operated under the strict quality assurance criteria of the German screening colonoscopy program. Because of the low CRC prevalence in a screening setting, additional CRC cases from the DACHSplus study were included. In the DACHSplus study, newly diagnosed CRC patients were recruited in 4 cooperating hospitals before their treatment.

Written informed consent was obtained from each study participant. Both studies were approved by the ethics committee of the Medical Faculty Heidelberg of the University of Heidelberg (BLITZ study (178/2005): 13 June 2005 and DACHSplus study (310/2001): 27 March 2006) and by the ethics committees of the State Chambers of Physicians of Baden-Wuerttemberg, Rhineland-Palatinate and Hesse. 

### 2.2. Sample and Data Collection

Study participants recruited from 2005 through 2010 were asked to fill a sample cup (60 mL) with feces from a single bowel movement without any dietary or medicinal restrictions before starting bowel preparation for colonoscopy (BLITZ) or surgery (DACHSplus). Furthermore, they were instructed to store the stool-filled cups in a freezer and bring them to their colonoscopy appointment (BLITZ) or hospital admission (DACHSplus). After arrival, the samples were kept frozen at −20 °C and transported after an initial fecal hemoglobin measurement on dry ice to the German Cancer Research Center (DKFZ) for final storage at −80 °C.

Colonoscopy and histology reports, as well as medical and histological surgery reports were collected, and relevant data were independently extracted by two medical data managers in a blinded manner (unaware of the FIT results).

### 2.3. Selection of Study Participants

Study participants, who were recruited from 2005 through 2010 and who provided a stool sample were eligible for this study (Figure 1). After excluding participants due to the criteria shown in Figure 1, 1667 BLITZ study participants (screening setting) and 94 clinical CRC cases (clinical setting) were eligible. From both studies, all advanced neoplasm cases (CRC or AA) with enough stool material for the evaluation of 9 FITs were included. One screen-detected CRC case with UICC stage 0 was excluded. Therefore, 15 CRC cases and 200 AA cases from BLITZ and 79 CRC cases from DACHSplus were finally included. For analysis of specificity, 300 randomly selected participants free of CRC and AA, who provided enough stool material, were included. The random selection was performed using the SURVEYSELECT procedure in SAS.

More than 95% of the FIT measurements were conducted in the context of a previously reported study [17]. For the 29 CRC cases included in the final analysis, parallel FIT measurements were conducted in the context of another previously reported study [18], in which two (Eurolyser FOB test, OC-Sensor) of the previous nine FITs were not evaluated. Therefore, for these two FITs, the final analysis is based on 65 CRC cases (15 CRC cases from BLITZ + 50 CRC cases from DACHSplus).

### 2.4. Laboratory Analysis

Detailed information about the FITs are shown in Table 1. FIT measurements were performed at the DKFZ in Heidelberg or in nearby located laboratories of the manufacturers as previously reported in detail [17,18]. Briefly, each FIT has a brand-specific fecal sampling tube, which is filled with a hemoglobin stabilizing buffer and contains an extricable serrated stick for the collection of a defined amount of feces (range: 9.5–20 mg). The stick was inserted multiple times into different areas of the stool sample until the serrations (which transfer the defined amount of feces) were completely filled with stool and then pushed back into the tube once. To ensure equal pre-analytic conditions, all FIT tubes were filled in parallel after thawing of stool samples and stored at room temperature (range: 17–25 °C) until parallel FIT measurements on the next day. The externally evaluated FIT tubes were packed in a temperature-isolated manner and immediately sent to the cooperating companies via a special courier service. Test calibrators and test controls were performed according to the manufacturers’ instructions. All test measurements were conducted in a blinded manner.

### 2.5. Statistical Analysis

All quantitative FIT measurements were converted to the same and directly comparable unit of µg Hb/g feces [19].

Sensitivities were calculated for CRC according to UICC stage, T status, tumor location (proximal colon (caecum, ascending colon, hepatic flexure, transverse colon, splenic flexure), distal colon (descending colon, sigmoid colon, rectosigmoid junction), and rectum), sex, age group (50–59, 60–69, 70–79 years), and BMI group (normal weight: 18.5–24.9 kg/m^2^; overweight: 25.0–29.9 kg/m^2^, obesity: ≥30 kg/m^2^). UICC stage definitions followed the AJCC Cancer Staging Manual (7th edition) and are provided in Appendix A. Sensitivities for the detection of AA were calculated according to size, villous/tubulovillous architecture, high-grade dysplasia, by location (same definitions as above, but participants with multiple AAs were excluded from the analysis by location only, because the AAs were distributed across different colon sections), number of AAs, sex, age group (as above), and BMI group (as above). Specificities were computed among participants without CRC and AA at screening colonoscopy.

Sensitivity and specificity estimates were computed at the cutoff values recommended by the manufacturers (Table 1). Differences in overall specificities between FITs at their original cutoffs may obscure potential FIT-specific differences in associations between sensitivity and the assessed variables. In order to enhance the comparability of results among the different FIT brands, we additionally calculated sensitivities at cutoffs adjusted to yield an equal overall specificity of 93% [17]. One of the tests, QuikRead go iFOBT, could not be included in this comparison, as the cutoff value could not be lowered below 15 µg/g due to the restricted analytical working range. 

The 95% confidence intervals (CIs) of sensitivities and specificities were calculated using the “exact” (Clopper–Pearson) method. The Cochran–Armitage Test for trend in proportions was used to evaluate the statistical significance of trends in sensitivities across T status, UICC stage, location, age, and BMI. Fisher’s exact test was used to test for differences in sensitivities between both sexes, AA size, architecture, dysplasia, and number of AAs. 

Statistical analyses were performed using SAS Enterprise Guide, version 7.1 (SAS Institute, Cary, NC, USA). Statistical significance was indicated by two-sided *p*-values below 0.05.

## 3. Results

### 3.1. Study Population

Characteristics of the study population are shown in Table 2. Screening (*n* = 15) and clinical (*n* = 79) CRC cases were combined. For two tests (Eurolyser FOB test, OC-Sensor), the evaluation of CRC sensitivity is based on 65 CRC cases overall. 

### 3.2. Sensitivities at Original Cutoffs

At original cutoff values, overall specificities ranged from 86% to 98% (median: 91%) across the nine FITs (Table 3).

Overall sensitivity for CRC ranged from 63% to 83% (median: 73%) (Figure 2A, Table 3). Sensitivities of stage I cancers ranged from 52% to 72% (median: 62%), whereas sensitivities for more advanced stages (II–IV) were consistently higher (by 8–25% points), ranging from 59% to 93% (median: 77%). However, results were not statistically significant. An even stronger gradient was observed according to T status. Sensitivities for T1 cancers ranged from 39% to 56% (median: 44%), whereas sensitivities for T2–T4 cancers were by 7–56% points higher, ranging from 57% to 100% (median: 80%). Sensitivity was the highest among T4 cancers (range: 78–100%, median: 89%). The trend towards higher sensitivities with more advanced T status was statistically significant (*p* < 0.05) among all seven FITs that evaluated the larger sample size (*n* = 94 cases).

Overall sensitivity for AA ranged from 18% to 44% (median: 31%) (Figure 2A, Table 3). Sensitivities were by 13–28% points higher for large AAs (≥1 cm) in comparison to small AAs, and statistical significance was found among all nine FITs (*p* < 0.05). Sensitivity of AA with high-grade dysplasia was detected at consistently higher sensitivities (by 7–16% points) than AAs without high-grade dysplasia; however, results were not statistically significant. Participants with multiple AAs (*n* = 33) yielded substantially higher sensitivities (range: 45–67%, median: 64%) than participants with only a single AA (range: 11–39%, median: 25%), and differences between both groups were statistically significant among all nine FITs (all *p*-values <0.01). Interestingly, for 7 of the 9 FITs, sensitivity for the detection of multiple AA was by 7–25% points higher than sensitivity for the detection of T1 cancers. 

Looking at the tumor location (Table 3), the highest sensitivity was consistently observed for rectum cancers, followed by proximal and distal colon cancers. Furthermore, sensitivities for CRC were consistently higher among men than among women. Younger individuals yielded generally higher sensitivities than older ones, and overweight or obese patients yielded generally higher sensitivities than normal-weighted individuals. However, none of these observed differences in sensitivities according to location, sex, age, and BMI reached statistical significance.

Furthermore, sensitivities for AA were consistently higher for males than for females, for older participants than for younger participants, and for obese participants than for over- or normal-weighted participants, even though most of these differences did not reach statistical significance (Table 3).

### 3.3. Sensitivities at Cutoffs Yielding 93% Overall Specificity

In general, overall and subgroup-specific sensitivities became very similar between the different FIT brands when cutoff values were adjusted to yield the same overall specificity (Figure 2B, Table 4). 

Very similar to the results obtained at original cutoffs, strong variations of sensitivity of CRC detection by tumor stage were seen (Figure 2B, Table 4). Sensitivities for stage I were by up to 27% points lower in comparison to more advanced stages II–IV; however, the results were not statistically significant for 7 of the 8 FITs. Again, an even stronger gradient in sensitivities was observed according to T status. Sensitivities were by 14–61% points lower for T1 cancers in comparison to T2–T4, and for 7 of the 8 FITs included in this analysis, trends towards higher sensitivities by T status were statistically significant (*p* < 0.05). 

Overall sensitivity for AA ranged from 27% to 32% (median: 30%) (Figure 2B, Table 4). Sensitivities were consistently higher for large AAs (≥1 cm) than for small AAs, and for 7 of the 8 FITs, these differences were statistically significant (*p* < 0.05). Furthermore, we found in an ancillary analysis that sensitivities for AA increased statistically significantly (*p* < 0.05) from <0.5 cm to ≥0.5–1 cm, ≥1–3 cm, and ≥3 cm (Appendix A). AA with high-grade dysplasia showed by up to 17% points higher sensitivities in comparison to AA without high-grade dysplasia; however, results were not statistically significant. Among participants with a single AA (*n* = 167), AAs located in the distal colon showed higher sensitivities than AAs located in the rectum or proximal colon, but a statistically significant difference was not found. Again, sensitivities were much higher (by 28–41% points) among participants with multiple AAs than among participants with a single AA, and this difference was statistically significant for all eight FITs (*p* < 0.005). Additionally, sensitivities for multiple AAs were again higher (by up to 25% points) for 7 of the 8 FITs than for T1 cancers. 

In line with results at original cutoffs, highest sensitivities were consistently observed for rectum cancers, whereas sensitivities were even slightly lower for cancers in the distal than for cancers in the proximal colon (Table 4). Furthermore, consistently higher sensitivities for CRC were found for men vs. women; generally slightly higher sensitivities for younger than for older individuals; and for overweight or obese patients, the sensitivities were slightly higher than for normal-weighted individuals. Again, however, none of the differences according to location, sex, age, and BMI reached statistical significance.

Furthermore, sensitivities for AA tended to be higher for men than for women even though differences did not reach statistical significance (Table 4). Sensitivities increased consistently from younger to older participants, and for half of the FITs, this trend towards higher sensitivities by age was statistically significant (*p* < 0.05).

## 4. Discussion

In this study, we evaluated for the first time the sensitivity for CRC and AA detection of a large number of different quantitative FITs according to tumor stage (overall stage and T status), histological characteristics of AA, location, sex, age, and BMI, using fecal samples of participants of screening colonoscopy enriched with newly diagnosed clinical CRC cases. Strong associations between more advanced tumor stage and higher sensitivity were observed. These associations were particularly strong and statistically significant by T status (differences between T1 and T2–T4 by up to ~60% points) but also notable among overall UICC stages (differences between stage I and II-IV by up to ~25% points). Participants with multiple AAs yielded only slightly lower sensitivities than those with UICC stage I cancer and by up to 25% points higher sensitivities than those with T1 cancers.

The observed gradient in sensitivities with increasing UICC stage was even stronger than the gradient estimated in a previous meta-analysis [9] where pooled sensitivity for more advanced cancer stage was by up to ~15% points higher than for stage I cancers. However, in a recent study [10] including 435 newly diagnosed CRC cases, a similarly strong difference in sensitivity by up to ~35% points between stage I and more advanced stages (II-IV) was observed. Furthermore, two previous studies [20,21] found an even stronger difference (by ~50% points) in sensitivity between stage I and more advanced cancer stages. With respect to T status, in the aforementioned meta-analysis [9], the observed difference between pooled sensitivity for T1 and T2–T4 cancers (by up to ~40% points) was also less pronounced than in our analysis (by up to ~60% points). However, FIT sensitivities varied widely across the included studies [9], which possibly affected the observed differences. Yet, Hirata et al. [22] and Kim et al. [23] also found a similarly strong difference in sensitivity between T1 and more advanced T statuses (by ~60–70% points), and in a recent publication [10], the difference between T1 and T2–T4 cancers ranged up to ~50% points. Furthermore, we found that sensitivity was the highest among T4 cancers. Even though larger tumors tend to bleed stronger, a previous study [10] found lower sensitivities in T4 compared to T3 cancers and hypothesized that clinically manifest anemia lowered FIT sensitivity in T4 compared to T3 tumors. Future studies might consider stool and blood hemoglobin levels to investigate this topic in detail. Previous studies suggested consistently higher sensitivities of FIT for distal CRC compared to proximal colon cancers [24]. However, those studies did not differentiate between tumors in the distal colon and tumors in the rectum. To our knowledge, only one previous study [10] reported sensitivities for distal colon cancer and rectal cancer separately. In our study, we evaluated for the first time the sensitivities for distal colon and rectal cancer separately for a large number of different FITs and found a consistently (albeit not statistically significantly) higher sensitivity for rectal cancer compared to distal colon cancers for all FITs, although there was no participant with T4 cancer among the rectal cancer cases. Interestingly, among participants with a single AA, sensitivities were highest for distal AA cases and lowest for proximal AA cases, although statistical significance was not reached. Future studies reporting on sensitivity according to location should thus consider additional stratification of distal advanced neoplasms into those located in the distal colon and rectum.

In line with findings from previous studies [5,25,26,27,28], we observed consistently (albeit in the vast majority not statistically significantly) higher sensitivities among men than among women across all nine FITs. A potential explanation for this observation might be the higher proportion of T1 and lower proportion of T4 cancers among women than among men (32% vs. 11% and 5% vs. 13%, respectively). For AA detection, men showed a higher proportion of multiple AAs (20% vs. 9%) and a higher proportion of large AAs (75% vs. 66%) compared to women. Future studies might consider investigating FIT accuracy in multivariate analysis if case numbers allow it. Moreover, we found lower sensitivities among elderly CRC participants, which goes in line with the results of Selby et al. [5] For the detection of AA, we found a non-statistically significant trend towards higher sensitivities with higher age, which has also been shown in previous studies [26,28] and might be explained by the rising proportion of multiple AAs with age (from 9% to 17% and 27% in age groups 50–59, 60–69, and 70–79 years, respectively). 

Our study has a number of strengths. To our knowledge, this is the first study to directly compare sensitivities of different FITs among the same study participants with additional consideration of tumor stage, AA characteristics, location, sex, age, and BMI. This increases comparability to other studies using only one FIT and precludes the potential influence of the different study designs, as an additional reason for varying sensitivities according to the different patient characteristics. Furthermore, we investigated the FITs at different cutoffs: the original cutoffs and at cutoffs adjusted to yield the same overall specificity of 93%. Sensitivities for all FITs were reported stratified by a range of potentially influential factors, such as tumor stage (overall UICC and T status) and location; number; size; histology; and location of AA, by age, sex, and BMI. Ours is also the first study to our knowledge that compared the sensitivity of T1 and UICC stage I cancers with sensitivity of AA with various characteristics and found that sensitivities for multiple AAs were particularly high and exceeded sensitivity for T1 cancers by up to 25% points. Furthermore, several exclusion criteria were applied to the recruited study participants: Of the screening participants, we excluded those not in the relevant age for screening (<50 or >79 years), those at elevated risk (IBD [29] or history of colorectal neoplasms), or decreased risk of CRC (colonoscopy in the past 5 years [30]), and participants with potentially inadequate gold-standard exam (colonoscopy) to verify FIT findings (incomplete colonoscopy or inadequate bowel cleansing). Of the participants recruited after confirmed CRC diagnosis, we also excluded those not in the relevant age for screening or not representative for average-risk participants who developed CRC (prior diagnosis of IBD or history of CRC), and those who had neoadjuvant therapy before stool collection, because chemotherapy might influence FIT results.

Our study also has limitations. Despite the large overall number of FIT measurements (*n* > 5000), the limited number of CRC and AA cases resulted in wide confidence intervals of subgroup-specific sensitivities and limited power for detecting differences across subgroups. Furthermore, stool samples were frozen and thawed before analysis, which differs from the recommended procedure of sample processing (transferring the fresh stool material directly into the fecal sampling tubes, which are filled with a hemoglobin stabilizing buffer and analyzing the FITs within a short period without any freezing and thawing). However, we found very similar diagnostic performance between both above-mentioned fecal sampling procedures [31] and repeated freezing and thawing as well as long-term frozen storage at −80 °C had only very little effect on measurable hemoglobin concentrations and resulting FIT performance [32]. 

## 5. Conclusions

In conclusion, this study found strong differences in sensitivity according to tumor stage, AA size, and number of AAs. Across different FITs, particularly low sensitivities were observed for T1 cancers, and sensitivities were even higher for multiple AAs than for T1 cancers. By contrast, differences in sensitivity were small across the FIT brands when comparing them at equal specificity. These findings warrant further research to investigate causes of false-negative FITs and to assess the potential for improving sensitivity for early-stage CRCs (UICC stage I) and T1 cancers in particular.

## Figures and Tables

**Figure 1 cancers-13-00644-f001:**
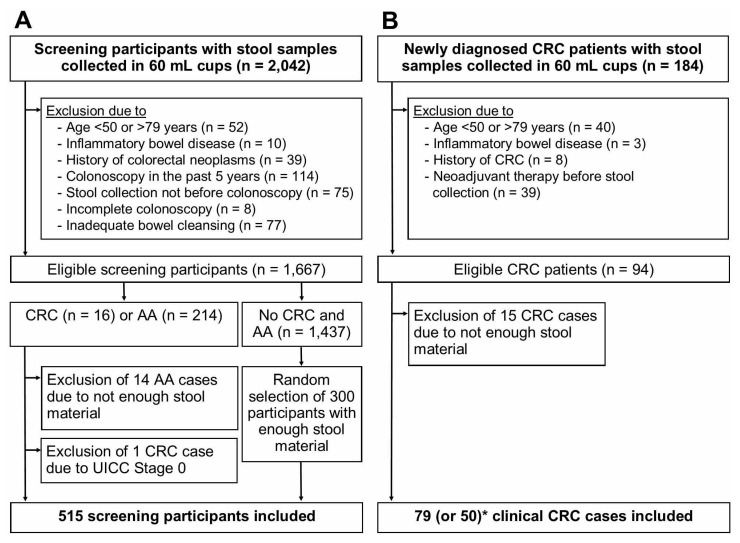
Flow diagram of study participants. (**A**), BLITZ study (screening setting); (**B**), DACHSplus study (clinical setting); * for two FITs (Eurolyser FOB test and OC-Sensor), only 50 CRC cases from DACHSplus were included in the final analysis. Abbreviations: AA, advanced adenoma; CRC, colorectal cancer.

**Figure 2 cancers-13-00644-f002:**
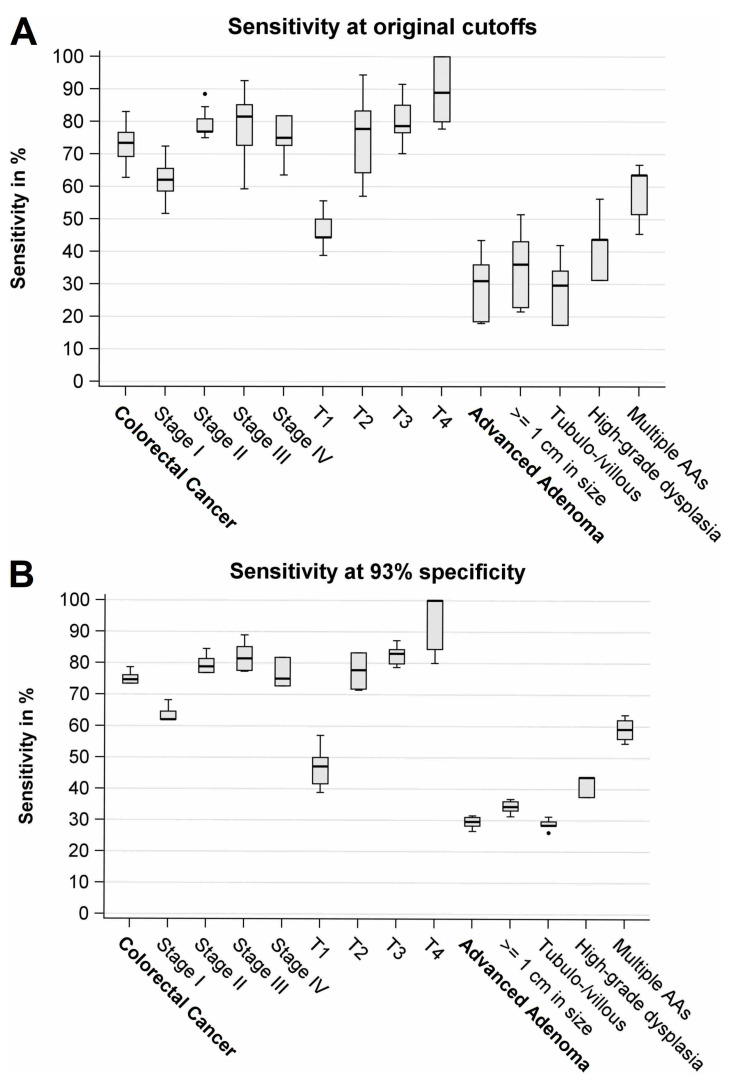
Box plots of sensitivities at original cutoffs (**A**) and at adjusted cutoffs (**B**) across the FITs. AA, Advanced adenoma.

**Table 1 cancers-13-00644-t001:** Characteristics of the nine fecal immunochemical tests (FITs) (sorted by specificity value at original cutoffs).

FIT Brand	Manufacturer, City, Country	Fecal Sampling Tube (Fecal Mass/Buffer Volume)	Analytical Instrument	Analytical Reading Technique	Analytical Working Range (µg Hb/g Feces)
IDK Hb ELISA	Immundiagnostik, Bensheim, Germany	IDK Extract (15 mg/1.5 mL)	DSX by Dynex Technologies	Enzyme-linked immunosorbent assay (ELISA)	0.086 to 50
QuantOn Hem	Immundiagnostik, Bensheim, Germany	QuantOn Hem TUBE (15 mg/1.5 mL)	QuantOn Hem test cassette and Smartphone *	Immunoaffinity chromatography and Photometry	0.3 to 100
immoCARE-C	CARE diagnostica, Möllersdorf, Austria	Sample Collection Tube (20 mg/2.5 mL)	immoCARE-C test cassette and CAREcube	Immunoaffinity chromatography and Photometry	3.75 to 250
RIDASCREEN Hb	R-Biopharm, Darmstadt, Germany	RIDA TUBE Hb (10 mg/2.5 mL)	DSX by Dynex Technologies	Enzyme-linked immunosorbent assay (ELISA)	0.65 to 50
CAREprime	Alfresa Pharma, Osaka, Japan	Specimen Collection Container A (9.5 mg/1.9 mL)	CAREprime	Immunoturbidimetry	1.6 to 240
SENTiFIT-FOB Gold	Sentinel Diagnostics, Milan, Italy	SENTiFIT pierceTube (10 mg/1.7 mL)	SENTiFIT 270 analyzer	Immunoturbidimetry	1.7 to 129.88
QuikRead go iFOBT	Orion Diagnostica, Espoo, Finland	QuikRead go iFOBT Sampling Set (10 mg/2.0 mL)	QuikRead go	Immunoturbidimetry	15 to 200
Eurolyser FOB test	Eurolyser Diagnostica, Salzburg, Austria	Eurolyser FOB Sample Collector (19.9 mg/1.6 mL)	Eurolyser CUBE	Immunoturbidimetry	2.01 to 80.4
OC-Sensor	Eiken Chemical, Tokyo, Japan	OC Auto-Sampling Bottle 3 (10 mg/2.0 mL)	OC Sensor io	Immunoturbidimetry	10 to 200

**Table 2 cancers-13-00644-t002:** Study population.

Screening and Clinical CRC Cases Combined (*n* (%))			Screening AA Cases (*n* (%))		Screening Participants Free of CRC and AA (*n* (%))	
**Total**	94 (100)	65 (100) *	**Total**	200 (100)	**Total**	300 (100)
**Sex**			**Sex**		**Sex**	
Women	38 (40.4)	26 (40.0)	Women	68 (34.0)	Women	155 (51.7)
Men	56 (59.6)	39 (60.0)	Men	132 (66.0)	Men	145 (48.3)
**Age (years)**			**Age (years)**		**Age (years)**	
50–59	15 (16.0)	13 (20.0)	50–59	67 (33.5)	50–59	119 (39.7)
60–69	42 (44.7)	28 (43.1)	60–69	88 (44.9)	60–69	140 (46.7)
70–79	37 (39.4)	24 (36.9)	70–79	45 (22.5)	70–79	41 (13.7)
**BMI (kg/m^2^)**			**BMI (kg/m^2^)**		**BMI (kg/m^2^)**	
<18.5 (underweight)	1 (1.1)	-	<18.5 (underweight)	2 (1.0)	<18.5 (underweight)	1 (0.3)
18.5 to <25 (normal)	31 (33.0)	21 (32.3)	18.5 to <25 (normal)	54 (27.0)	18.5 to <25 (normal)	105 (35.0)
25 to <30 (overweight)	42 (44.7)	31 (47.7)	25 to <30 (overweight)	89 (44.5)	25 to <30 (overweight)	124 (41.3)
≥30 (obesity)	18 (19.2)	11 (16.9)	≥30 (obesity)	51 (25.5)	≥30 (obesity)	64 (21.3)
Missing	2 (2.1)	2 (3.1)	Missing	4 (2.0)	Missing	6 (2.0)
**Location**			**Location**			
Proximal colon	32 (34.0)	22 (33.9)	Proximal colon	52 (26.0)		
Distal colon	30 (31.9)	22 (33.9)	Distal colon	70 (35.0)		
Rectum	31 (33.0)	20 (30.8)	Rectum	45 (22.5)		
Missing	1 (1.1)	1 (1.5)	Multiple AAs	33 (16.5)		
**UICC Stage**			**Size ≥ 1 cm**			
I	29 (30.9)	22 (33.9)	No	56 (28.0)		
II	26 (27.7)	16 (24.6)	Yes	144 (72.0)		
III	27 (28.7)	22 (33.9)				
IV	11 (11.7)	4 (6.2)	**Tubulovillous/villous**			
Missing	1 (1.1)	1 (1.5)	No	62 (31.0)		
			Yes	138 (69.0)		
**T status**						
T1	18 (19.2)	14 (21.5)	**High-grade dysplasia**			
T2	18 (19.2)	14 (21.5)	No	184 (92.0)		
T3	47 (50.0)	32 (49.2)	Yes	16 (8.0)		
T4	9 (9.6)	5 (7.7)				
Missing	2 (2.1)	-	**Number of AAs**			
			Single AA	167 (83.5)		
			Multiple AAs	33 (16.5)		

* For two FITs (Eurolyser FOB test and OC-Sensor) the analyses for detection of CRC were based on 65 CRC cases. Abbreviations: AA, advanced adenoma; BMI, body mass index; CRC, Colorectal cancer; UICC, Union for International Cancer Control.

**Table 3 cancers-13-00644-t003:** Sensitivity (% (95%CI)) at original cutoffs.

Variable	IDK Hb ELISA	QuantOn Hem	immoCARE-C	RIDASCREEN Hb	CAREprime	SENTiFIT-FOB Gold	QuikRead go iFOBT	Eurolyser FOB test	OC-Sensor
**Cutoff**	2.0 µg/g	3.7 µg/g	6.25 µg/g	8.0 µg/g	6.3 µg/g	17.0 µg/g	15.0 µg/g	8.04 µg/g	10 µg/g
**Specificity**	85.7 (81.2–89.4)	85.7 (81.2–89.4)	90.0 (86.0–93.2)	90.7 (86.8–93.7)	91.3 (87.6–94.3)	96.3 (94.2–98.2)	96.7 (94.0–98.4)	97.0 (94.4–98.6)	97.7 (95.3–99.1)
	**Sensitivity for CRC detection**								
**Overall**	83.0 (73.8–90.0)	83.0 (73.8–90.0)	76.6 (66.7–84.7)	74.5 (64.4–82.9)	73.4 (63.3–82.0)	70.2 (79.2–62.8)	62.8 (52.2–72.5)	66.2 (53.4–77.4) *	69.2 (56.6–80.1) *
***by UICC stage***									
I	72.4 (52.8–87.3)	69.0 (49.2–84.7)	65.5 (45.7–82.1)	62.1 (42.3–79.3)	62.1 (42.3–79.3)	58.6 (38.9–76.5)	51.7 (32.5–70.6)	54.6 (32.2–75.6)	59.1 (36.–79.3)
II	84.6 (65.1–95.6)	88.5 (69.9–97.6)	80.8 (60.7–93.5)	76.9 (56.4–91.0)	76.9 (56.4–91.0)	76.9 (56.4–91.0)	76.9 (56.4–91.0)	75.0 (47.6–92.7)	75.0 (47.6–92.7)
III	92.6 (75.7–99.1)	92.6 (75.7–99.1)	81.5 (61.9–93.7)	85.2 (66.3–95.8)	81.5 (61.9–93.7)	74.1 (53.7–88.9)	59.3 (38.8–77.6)	68.2 (45.1–86.1)	72.7 (49.8–89.3)
IV	81.8 (48.2–97.7)	81.8 (48.2–97.7)	81.8 (48.2–97.7)	72.7 (39.0–94.0)	72.7 (39.0–94.0)	72.7 (39.0–94.0)	63.6 (30.8–89.1)	75.0 (19.4–99.4)	75.0 (19.4–99.4)
*p* (trend)	0.1785	0.1054	0.1897	0.1649	0.2526	0.2716	0.6030	0.3438	0.4048
***by T stage***									
T1	55.6 (30.8–78.5)	55.6 (30.8–78.5)	38.9 (17.3–64.3)	44.4 (21.5–69.2)	44.4 (21.5–69.2)	44.4 (21.5–69.2)	44.4 (21.5–69.2)	50.0 (23.0–77.0)	50.0 (23.0–77.0)
T2	94.4 (72.7–99.9)	83.3 (58.6–96.4)	88.9 (65.3–98.6)	83.3 (58.6–96.4)	77.8 (52.4–93.6)	72.2 (46.5–90.3)	61.1 (35.8–82.7)	57.1 (28.9–82.3)	64.3 (35.1–87.2)
T3	87.2 (74.3–95.2)	91.5 (79.6–97.6)	85.1 (71.7–93.8)	78.7 (64.3–89.3)	78.7 (64.3–89.3)	76.6 (62.0–87.7)	70.2 (55.1–82.7)	75.0 (56.6–88.5)	78.1 (60.0–90.7)
T4	100 (66.4–100)	100 (66.4–100)	88.9 (51.8–99.7)	100 (66.4–100)	100 (66.4–100)	88.9 (51.8–99.7)	77.8 (40.0–97.2)	80.0 (28.4–99.5)	80.0 (28.4–99.5)
*p* (trend)	**0.0033**	**0.0004**	**0.0006**	**0.0023**	**0.0017**	**0.0083**	**0.0437**	0.0856	0.0573
***by location***									
Proximal only	81.3 (63.6–92.8)	81.3 (63.6–92.8)	75.0 (56.6–88.5)	75.0 (56.6–88.5)	71.9 (53.3–86.3)	65.6 (46.8–81.4)	56.3 (37.7–73.6)	63.6 (40.7–82.8)	63.6 (40.7–82.8)
Distal only	76.7 (57.7–90.1)	76.7 (57.7–90.1)	66.7 (47.2–82.7)	66.7 (47.2–82.7)	70.0 (50.6–85.3)	63.3 (43.9–80.1)	56.7 (37.4–74.5)	50.0 (28.2–71.8)	54.6 (32.2–75.6)
Rectum only	90.3 (74.3–98.0)	90.3 (74.3–98.0)	87.1 (70.2–96.4)	80.7 (62.5–92.6)	77.4 (58.9–90.4)	80.7 (62.5–92.6)	74.2 (55.4–88.1)	85.0 (62.1–96.8)	90.0 (68.3–98.8)
*p* (trend)	0.4089	0.4089	0.3037	0.6692	0.6733	0.2204	0.1563	0.1973	0.0974
***by sex***									
Women	73.7 (56.9–86.6)	79.0 (62.7–90.5)	68.4 (51.4–82.5)	65.8 (48.7–80.4)	68.4 (51.4–82.5)	63.2 (46.0–78.2)	55.3 (38.3–71.4)	57.7 (36.9–76.7)	61.5 (40.6–79.8)
Men	89.3 (78.1–96.0)	85.7 (73.8–93.6)	82.1 (69.6–91.1)	80.4 (67.6–89.8)	76.8 (63.6–87.0)	75.0 (61.6–85.6)	67.9 (54.0–79.7)	71.8 (55.1–85.0)	74.4 (57.9–87.0)
*p*-value	0.0563	0.4146	0.1422	0.1488	0.4762	0.2547	0.2778	0.2898	0.2885
***by age (years)***									
50–59	86.7 (59.5–98.3)	93.3 (68.1–99.8)	86.7 (59.5–98.3)	80.0 (51.9–95.7)	73.3 (44.9–92.2)	80.0 (51.9–95.7)	73.3 (44.9–92.2)	76.9 (46.2–95.0)	76.9 (46.2–95.0)
60–69	83.3 (68.6–93.0)	81.0 (65.9–91.4)	73.8 (58.0–86.1)	73.8 (58.0–86.1)	73.8 (58.0–86.1)	69.1 (52.9–82.4)	61.9 (45.6–76.4)	57.1 (37.2–75.5)	64.3 (44.1–81.4)
70–79	81.1 (64.8–92.0)	81.1 (64.8–92.0)	75.7 (58.8–88.2)	73.0 (55.9–86.2)	73.0 (55.9–86.2)	67.6 (50.2–82.0)	59.5 (42.1–75.3)	70.8 (48.9–87.4)	70.8 (48.9–87.4)
*p* (trend)	0.7022	0.4435	0.6084	0.7400	1.00	0.5260	0.4538	1.00	0.8582
***by BMI***									
Normal	80.7 (62.5–92.6)	83.9 (66.3–94.6)	74.2 (55.4–88.1)	71.0 (52.0–85.8)	71.0 (52.0–85.8)	64.5 (45.4–80.8)	54.8 (36.0–72.7)	61.9 (38.4–81.9)	66.7 (43.0–85.4)
Overweight	88.1 (74.4–96.0)	81.0 (65.9–91.4)	81.0 (65.9–91.4)	81.0 (65.9–91.4)	78.6 (63.2–89.7)	76.2 (60.6–88.0)	69.1 (52.9–82.4)	64.5 (45.4–80.8)	67.7 (48.6–83.3)
Obesity	83.3 (58.6–96.4)	88.9 (65.3–98.6)	77.8 (52.4–93.6)	72.2 (46.5–90.3)	72.2 (46.5–90.3)	72.2 (46.5–90.3)	66.7 (41.0–86.7)	81.8 (48.2–97.7)	81.8 (48.2–97.7)
*p* (trend)	0.8417	0.8463	0.7298	0.8663	0.8688	0.5233	0.3675	0.3447	0.5576
	**Sensitivity for AA detection**								
**Overall**	43.5 (36.5–50.7)	41.5 (34.6–48.7)	35.2 (28.6–42.2) **	36.0 (29.4–43.1)	31.0 (24.7–37.9)	18.0 (12.9–24.0)	18.5 (13.4–24.6)	19.5 (14.3–25.7)	18.0 (12.9–24.0)
***by size ≥ 1 cm***									
No	23.2 (13.0–36.4)	26.8 (15.8–40.3)	23.2 (13.0–36.4)	17.9 (8.9–30.4)	17.9 (8.9–30.4)	8.9 (3.0–19.6)	8.9 (3.0–19.6)	5.4 (1.1–14.9)	5.4 (1.1–14.9)
Yes	51.4 (42.9–59.8)	47.2 (38.9–55.7)	39.9 (31.8–48.4) **	43.1 (34.8–51.6)	36.1 (28.3–44.5)	21.5 (15.1–29.1)	22.2 (15.7–29.9)	25.0 (18.2–32.9)	22.9 (16.3–30.7)
*p-value*	**0.0004**	**0.0104**	**0.0319**	**0.0009**	**0.0165**	**0.0410**	**0.0411**	**0.0012**	**0.0035**
**by tubulo/-villous histology**									
No	46.8 (34.0–59.9)	43.6 (31.0–56.7)	38.7 (26.6–51.9)	40.3 (28.1–53.6)	33.9 (22.3–47.0)	19.4 (10.4–31.4)	19.4 (10.4–31.4)	24.2 (14.2–36.7)	19.4 (10.4–31.4)
Yes	42.0 (33.7–50.7)	40.6 (32.3–49.3)	33.6 (25.7–42.1) **	34.1 (26.2–42.6)	29.7 (22.2–38.1)	17.4 (11.5–24.8)	18.1 (12.1–25.6)	17.4 (11.5–24.8)	17.4 (11.5–24.8)
*p-value*	0.5414	0.7569	0.5231	0.4278	0.6206	0.8425	0.8455	0.3344	0.8425
**by high-grade dysplasia**									
No	42.9 (35.7–50.4)	40.2 (33.1–47.7)	34.4 (27.6–41.8) **	35.3 (28.4–42.7)	29.9 (23.4–37.1)	16.9 (11.7–23.1)	17.4 (12.2–23.7)	18.5 (13.2–24.9)	16.9 (11.7–23.1)
Yes	50.0 (24.7–75.4)	56.3 (29.9–80.3)	43.8 (19.8–70.1)	43.8 (19.8–70.1)	43.8 (19.8–70.1)	31.3 (11.0–58.7)	31.3 (11.0–58.7)	31.3 (11.0–58.7)	31.3 (11.0–58.7)
*p-value*	0.6084	0.2903	0.5859	0.5892	0.2676	0.1732	0.1830	0.2053	0.1732
***by location***									
Proximal only	34.6 (22.0–49.1)	30.8 (18.7–45.1)	23.1 (12.5–36.8)	25.0 (14.0–39.0)	19.2 (9.6–32.5)	7.7 (2.1–18.5)	11.5 (4.4–23.4)	11.5 (4.4–23.4)	9.6 (3.2–21.0)
Distal only	48.6 (36.4–60.8)	44.3 (32.4–56.7)	36.2 (25.0–48.7) **	37.1 (25.9–49.5)	30.0 (19.6–42.1)	14.3 (7.1–24.7)	14.3 (7.1–24.7)	14.3 (7.1–24.7)	17.1 (9.2–28.0)
Rectum only	28.9 (16.4–44.3)	31.1 (18.2–46.7)	26.7 (14.6–41.9)	26.7 (14.6–41.9)	22.2 (11.2–37.1)	8.9 (2.5–21.2)	8.9 (2.5–21.2)	11.1 (3.7–24.1)	8.9 (2.5–21.2)
*p (trend)*	0.6781	0.9166	0.6584	0.8265	0.7247	0.8711	0.7572	1.00	1.00
***by number of AAs***									
Single	38.9 (31.5–46.8)	36.5 (29.2–44.3)	29.5 (22.7–37.1) **	30.5 (23.7–38.1)	24.6 (18.2–31.8)	10.8 (6.5–16.5)	12.0 (7.5–17.9)	14.4 (9.4–20.6)	12.6 (8.0–18.6)
Multiple	66.7 (48.2–82.0)	66.7 (48.2–82.0)	63.6 (45.1–79.6)	63.6 (45.1–79.6)	63.6 (45.1–79.6)	54.6 (36.4–71.9)	51.5 (33.5–69.2)	45.5 (28.1–63.7)	45.5 (28.1–63.7)
*p-value*	**0.0040**	**0.0018**	**0.0003**	**0.0006**	**<0.0001**	**<0.0001**	**<0.0001**	**0.0002**	**<0.0001**
***by sex***									
Women	36.8 (25.4–49.3)	30.9 (20.2–43.3)	29.4 (19.0–41.7)	28.0 (17.7–40.2)	23.5 (14.1–35.4)	13.2 (6.2–23.6)	11.8 (5.2–21.9)	14.7 (7.3–25.4)	13.2 (6.2–23.6)
Men	47.0 (38.2–55.9)	47.0 (38.2–55.9)	38.2 (29.8–47.1) **	40.2 (31.7–49.0)	34.9 (26.8–43.6)	20.5 (13.9–28.4)	22.0 (15.2–30.0)	22.0 (15.2–30.0)	20.5 (13.9–28.4)
*p-value*	0.1788	**0.0341**	0.2735	0.1196	0.1094	0.2468	0.0865	0.2609	0.2468
***by age (years)***									
50–59	34.3 (23.2–46.9)	35.8 (24.5–48.5)	28.8 (18.3–41.3) **	28.4 (18.0–40.7)	28.4 (18.0–40.7)	13.4 (6.3–24.0)	14.9 (7.4–25.7)	13.4 (6.3–24.0)	11.9 (5.3–22.2)
60–69	48.9 (38.1–59.8)	42.1 (31.6–53.1)	34.1 (24.3–45.0)	35.2 (25.3–46.1)	27.3 (18.3–37.8)	18.2 (10.8–27.8)	19.3 (11.7–29.1)	20.5 (12.6–30.4)	18.2 (10.8–27.8)
70–79	46.7 (31.7–62.1)	48.9 (33.7–64.2)	46.7 (31.7–62.1)	48.9 (33.7–64.2)	42.2 (27.7–57.9)	24.4 (12.9–39.5)	22.2 (11.2–37.1)	26.7 (14.6–41.9)	26.7 (14.6–41.9)
*p (trend)*	0.1505	0.1766	0.0710	**0.0366**	0.1805	0.1721	0.3288	0.0919	0.0620
***by BMI***									
Normal	37.0 (24.3–51.3)	35.2 (22.7–49.4)	33.3 (21.1–47.5)	29.6 (18.0–43.6)	31.5 (19.5–45.6)	16.7 (7.9–29.3)	16.7 (7.9–29.3)	16.7 (7.9–29.3)	16.7 (7.9–29.3)
Overweight	41.6 (31.2–52.5)	37.1 (27.1–48.0)	34.8 (25.0–45.7)	37.1 (27.1–48.0)	31.5 (22.0–42.2)	14.6 (8.0–23.7)	15.7 (8.9–25.0)	19.1 (11.5–28.8)	16.9 (9.8–26.3)
Obesity	54.9 (40.3–68.9)	58.8 (44.2–72.4)	40.0 (26.4–54.8) **	43.1 (29.4–57.8)	33.3 (20.8–47.9)	27.5 (15.9–41.7)	27.5 (15.9–41.7)	25.5 (14.3–39.6)	23.5 (12.8–37.5)
*p (trend)*	0.0771	**0.0177**	0.5405	0.1581	0.9169	0.1706	0.1749	0.2765	0.3839

* Analysis based on 65 CRC cases only. ** Analysis based on one less AA case. Abbreviations: AA, advanced adenoma; BMI, body mass index; CI, confidence interval; CRC colorectal cancer; UICC, Union for International Cancer Control. In bold statistical significant results.

**Table 4 cancers-13-00644-t004:** Sensitivity (% (95%CI)) at adjusted cutoffs yielding 93% specificity.

Variable	IDK Hb ELISA	QuantOn Hem	immoCARE-C	RIDASCREEN Hb	CAREprime	SENTiFIT-FOB Gold	Eurolyser FOB Test	OC-Sensor
**Cutoff**	4.8 µg/g	9.59 µg/g	9.2 µg/g	12.27 µg/g	6.65 µg/g	1.7 µg/g	2.01 µg/g	3.6 µg/g
**Specificity**	93.0 (89.5–95.6)	93.0 (89.5–95.6)	93.0 (89.5–95.6)	93.0 (89.5–95.6)	93.0 (89.5–95.6)	93.3 (89.9–95.9)	93.0 (89.5–95.6)	93.0 (89.5–95.6)
	**Sensitivity for CRC detection**							
**Overall**	76.6 (66.7–84.7)	78.7 (69.1–86.5)	73.4 (63.3–82.0)	73.4 (63.3–82.0)	73.4 (63.3–82.0)	75.5 (65.6–83.8)	75.4 (63.1–85.2) *	73.9 (61.5–84.0) *
***by UICC stage***								
I	62.1 (42.3–79.3)	65.5 (45.7–82.1)	62.1 (42.3–79.3)	62.1 (42.3–79.3)	62.1 (42.3–79.3)	62.1 (42.3–79.3)	68.2 (45.1–86.1)	63.6 (40.7–82.8)
II	76.9 (56.4–91.0)	84.6 (65.1–95.6)	80.8 (60.7–93.5)	76.9 (56.4–91.0)	76.9 (56.4–91.0)	76.9 (56.4–91.0)	81.3 (54.4–96.0)	81.3 (54.4–96.0)
III	88.9 (70.8–97.7)	85.2 (66.3–95.8)	77.8 (57.7–91.4)	81.5 (61.9–93.7)	81.3 (54.4–96.0)	85.2 (66.3–95.8)	77.3 (54.6–92.2)	77.3 (54.6–92.2)
IV	81.8 (48.2–97.7)	81.8 (48.2–97.7)	72.7 (39.0–94.0)	72.7 (39.0–94.0)	72.7 (39.0–94.0)	81.8 (48.2–97.7)	75.0 (19.4–99.4)	75.0 (19.4–99.4)
*p (trend)*	**0.0412**	0.1380	0.3607	0.2526	0.2526	0.0764	0.6565	0.3854
***by T status***								
T1	44.4 (21.5–69.2)	50.0 (26.0–74.0)	38.9 (17.3–64.3)	38.9 (17.3–64.3)	44.4 (21.5–69.2)	50.0 (26.0–74.0)	57.1 (28.9–82.3)	50.0 (23.0–77.0)
T2	83.3 (58.6–96.4)	77.8 (52.4–93.6)	83.3 (58.6–96.4)	83.3 (58.6–96.4)	77.8 (52.4–93.6)	72.2 (46.5–90.3)	71.4 (41.9–91.6)	71.4 (41.9–91.6)
T3	83.0 (69.2–92.4)	87.2 (74.3–95.2)	80.9 (66.7–90.9)	78.7 (64.3–89.3)	78.7 (64.3–89.3)	83.0 (69.2–92.4)	84.4 (67.2–94.7)	84.4 (67.2–94.7)
T4	100 (66.4–100)	100 (66.4–100)	88.9 (51.8–99.7)	100 (66.4–100)	100 (66.4–100)	100 (66.4–100)	80.0 (28.4–99.5)	80.0 (28.4–99.5)
*p (trend)*	**0.0006**	**0.0004**	**0.0017**	**0.0006**	**0.0017**	**0.0012**	0.0833	**0.0300**
***by location***								
Proximal only	78.1 (60.0–90.7)	78.1 (60.0–90.7)	68.8 (50.0–83.9)	71.9 (53.3–86.3)	71.9 (53.3–86.3)	71.9 (53.3–86.3)	63.6 (40.7–82.8)	68.2 (45.1–86.1)
Distal only	66.7 (47.2–82.7)	70.0 (50.6–85.3)	66.7 (47.2–82.7)	66.7 (47.2–82.7)	70.0 (50.6–85.3)	66.7 (47.2–82.7)	72.7 (49.8–89.3)	63.6 (40.7–82.8)
Rectum only	83.9 (66.3–94.6)	87.1 (70.2–96.4)	83.9 (66.3–94.6)	80.7 (62.5–92.6)	77.4 (58.9–90.4)	87.1 (70.2–96.4)	90.0 (68.3–98.8)	90.0 (68.3–98.8)
*p (trend)*	0.6600	0.4479	0.2045	0.4819	0.6733	0.1921	0.0752	0.1646
***by sex***								
Women	68.4 (51.4–82.5)	76.3 (59.8–88.6)	65.8 (48.7–80.4)	65.8 (48.7–80.4)	68.4 (51.4–82.5)	71.1 (54.1–84.6)	69.2 (48.2–85.7)	69.2 (48.2–85.7)
Men	82.1 (69.6–91.1)	80.4 (67.6–89.8)	78.6 (65.6–88.4)	78.6 (65.6–88.4)	76.8 (63.6–87.0)	78.6 (65.6–88.4)	79.5 (63.5–90.7)	76.9 (60.7–88.9)
*p-value*	0.1422	0.7979	0.2343	0.2343	0.4762	0.4673	0.3889	0.5695
***by age (years)***								
50–59	80.0 (51.9–95.7)	86.7 (59.5–98.3)	80.0 (51.9–95.7)	80.0 (51.9–95.7)	73.3 (44.9–92.2)	80.0 (51.9–95.7)	76.9 (46.2–95.0)	76.9 (46.2–95.0)
60–69	73.8 (58.0–86.1)	73.8 (58.0–86.1)	73.8 (58.0–86.1)	71.4 (55.4–84.3)	73.8 (58.0–86.1)	73.8 (58.0–86.1)	75.0 (55.1–89.3)	71.4 (51.3–86.8)
70–79	78.4 (61.8–90.2)	81.1 (64.8–92.0)	70.3 (53.0–84.1)	73.0 (55.9–86.2)	73.0 (55.9–86.2)	75.7 (58.8–88.2)	75.0 (53.3–90.2)	75.0 (53.3–90.2)
*p (trend)*	1.00	1.00	0.5151	0.7451	1.00	0.8677	1.00	1.00
***by BMI***								
Normal	74.2 (55.4–88.1)	80.7 (62.5–92.6)	67.7 (48.6–83.3)	71.0 (52.0–85.8)	71.0 (52.0–85.8)	74.2 (55.4–88.1)	66.7 (43.0–85.4)	71.4 (47.8–88.7)
Overweight	81.0 (65.9–91.4)	78.6 (63.2–89.7)	81.0 (65.9–91.4)	78.6 (63.2–89.7)	78.6 (63.2–89.7)	78.6 (63.2–89.7)	80.7 (62.5–92.6)	74.2 (55.4–88.1)
Obesity	77.8 (52.4–93.6)	83.3 (58.6–96.4)	72.2 (46.5–90.3)	72.2 (46.5–90.3)	72.2 (46.5–90.3)	77.8 (52.4–93.6)	81.8 (48.2–97.7)	81.8 (48.2–97.7)
*p (trend)*	0.7298	1.00	0.6199	0.8688	0.8688	0.8639	0.2981	0.6806
	**Sensitivity for AA detection**							
**Overall**	31.5 (25.1–38.4)	28.0 (21.9–34.8)	29.7 (23.4–36.5) **	31.0 (24.7–37.9)	29.5 (23.3–36.3)	28.5 (22.4–35.3)	31.0 (24.7–37.9)	26.5 (20.5–33.2)
***by size ≥ 1 cm***								
No	17.9 (8.9–30.4)	12.5 (5.2–24.1)	19.6 (10.2–32.4)	16.1 (7.6–28.3)	16.1 (7.6–28.3)	17.9 (8.9–30.4)	19.6 (10.2–32.4)	14.3 (6.4–26.2)
Yes	36.8 (28.9–45.2)	34.0 (26.4–42.4)	33.6 (25.9–41.9) **	36.8 (28.9–45.2)	34.7 (27.0–43.1)	32.6 (25.1–40.9)	35.4 (27.6–43.8)	31.3 (23.8–39.5)
*p-value*	**0.0108**	**0.0026**	0.0589	**0.0040**	**0.0096**	**0.0383**	**0.0404**	**0.0196**
**by tubulo/-villous histology**								
No	32.3 (20.9–45.3)	24.2 (14.2–36.7)	32.3 (20.9–45.3)	37.1 (25.2–50.3)	32.3 (20.9–45.3)	29.0 (18.2–42.0)	33.9 (22.3–47.0)	27.4 (16.9–40.2)
Yes	31.2 (23.6–39.6)	29.7 (22.2–38.1)	28.5 (21.1–36.8) **	28.3 (20.9–36.6)	28.3 (20.9–36.6)	28.3 (20.9–36.6)	29.7 (22.2–38.1)	26.1 (19.0–34.2)
*p-value*	0.8709	0.4971	0.6173	0.2478	0.6162	1.00	0.6206	0.8635
***by high-grade dysplasia***								
No	30.4 (23.9–37.6)	26.6 (20.4–33.6)	29.0 (22.5–36.1) **	29.9 (23.4–37.1)	28.3 (21.9–35.4)	27.2 (20.9–34.2)	30.4 (23.9–37.6)	25.5 (19.4–32.5)
Yes	43.8 (19.8–70.1)	43.8 (19.8–70.1)	37.5 (15.2–64.6)	43.8 (19.8–70.1)	43.8 (19.8–70.1)	43.8 (19.8–70.1)	37.5 (15.2–64.6)	37.5 (15.2–64.6)
*p-value*	0.2746	0.1545	0.5688	0.2676	0.2516	0.1618	0.5792	0.3746
***by location***								
Proximal only	15.4 (6.9–28.1)	15.4 (6.9–28.1)	15.4 (6.9–28.1)	17.3 (8.2–30.3)	17.3 (8.2–30.3)	17.3 (8.2–30.3)	19.2 (9.6–32.5)	13.5 (5.6–25.8)
Distal only	32.9 (22.1–45.1)	31.4 (20.9–43.6)	30.4 (19.9–42.7) **	31.4 (20.9–43.6)	28.6 (18.4–40.6)	28.6 (18.4–40.6)	32.9 (22.1–45.1)	25.7 (16.0–37.6)
Rectum only	26.7 (14.6–41.9)	17.8 (8.0–32.1)	24.4 (12.9–39.5)	22.2 (11.2–37.1)	20.0 (9.6–34.6)	20.0 (9.6–34.6)	24.4 (12.9–39.5)	17.8 (8.0–32.1)
*p (trend)*	0.2024	0.7185	0.2874	0.5572	0.7185	0.7185	0.5657	0.6113
***by number of AAs***								
Single	25.8 (19.3–33.1)	22.8 (16.6–29.9)	24.1 (17.8–31.3) **	24.6 (18.2–31.8)	22.8 (16.6–29.9)	22.8 (16.6–29.9)	26.4 (19.8–33.7)	19.8 (14.0–26.6)
Multiple	60.6 (42.1–77.1)	54.6 (36.4–71.9)	57.6 (39.2–74.5)	63.6 (45.1–79.6)	63.6 (45.1–79.6)	57.6 (39.2–74.5)	54.6 (36.4–71.9)	60.6 (42.1–77.1)
*p-value*	**0.0002**	**0.0005**	**0.0003**	**<0.0001**	**<0.0001**	**0.0002**	**0.0033**	**<0.0001**
***by sex***								
Women	22.1 (12.9–33.8)	25.0 (15.3–37.0)	26.5 (16.5–38.6)	22.1 (12.9–33.8)	20.6 (11.7–32.1)	20.6 (11.7–32.1)	22.1 (12.9–33.8)	20.6 (11.7–32.1)
Men	36.4 (28.2–45.2)	29.6 (21.9–38.1)	31.3 (23.5–40.0) **	35.6 (27.5–44.4)	34.1 (26.1–42.8)	32.6 (24.7–41.3)	35.6 (27.5–44.4)	29.6 (21.9–38.1)
*p-value*	0.0533	0.6182	0.5164	0.0542	0.0511	0.0979	0.0542	0.2362
***by age (years)***								
50–59	25.4 (15.5–37.5)	22.4 (13.1–34.2)	21.2 (12.1–33.0) **	25.4 (15.5–37.5)	25.4 (15.5–37.5)	20.9 (11.9–32.6)	25.4 (15.5–37.5)	20.9 (11.9–32.6)
60–69	31.8 (22.3–42.6)	28.4 (19.3–39.0)	30.7 (21.3–41.4)	28.4 (19.3–39.0)	27.3 (18.3–37.8)	27.3 (18.3–37.8)	31.8 (22.3–42.6)	23.9 (15.4–34.1)
70–79	40.0 (25.7–55.7)	35.6 (21.9–51.2)	40.0 (25.7–55.7)	44.4 (29.6–60.0)	40.0 (25.7–55.7)	42.2 (27.7–57.9)	37.8 (23.8–53.5)	40.0 (25.7–55.7)
*p (trend)*	0.1235	0.1379	**0.0360**	**0.0499**	0.1432	**0.0199**	0.1805	**0.0398**
***by BMI***								
Normal	27.8 (16.5–41.6)	25.9 (15.0–39.7)	29.6 (18.0–43.6)	27.8 (16.5–41.6)	27.8 (16.5–41.6)	27.8 (16.5–41.6)	27.8 (16.5–41.6)	25.9 (15.0–39.7)
Overweight	31.5 (22.0–42.2)	23.6 (15.2–33.8)	28.1 (19.1–38.6)	33.7 (24.0–44.5)	30.3 (21.0–41.0)	28.1 (19.1–38.6)	33.7 (24.0–44.5)	27.0 (18.1–37.4)
Obesity	37.3 (24.1–52.0)	39.2 (25.8–53.9)	34.0 (21.2–48.8) **	33.3 (20.8–47.9)	33.3 (20.8–47.9)	31.4 (19.1–45.9)	31.4 (19.1–45.9)	29.4 (17.5–43.8)
*p (trend)*	0.3476	0.1601	0.6704	0.6020	0.5971	0.7476	0.7534	0.7436

* Analysis based on 65 CRC cases only. ** Analysis based on one less AA case. QuikRead go iFOBT was excluded from the analysis at 93% specificity, because the cutoff value could not be set below 15 µg/g. Abbreviations: AA, advanced adenoma; BMI, body mass index; CI, confidence interval; CRC, colorectal cancer; UICC, Union for International Cancer Control. In bold statistical significant results.

## Data Availability

The data presented in this study are not publicly available, but data transfers with scientists signing a data transfer agreement for a specific research project can be granted by the last author.

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
