# Peer review of "Fecal Immunochemical Tests Detect Screening Participants with Multiple Advanced Adenomas Better than T1 Colorectal Cancers"

_cancers, 2021, doi:10.3390/cancers13040644_

Round 1

Reviewer 1 Report

To the authors/

Authors should be commended for their effort to improve early-stage CRC diagnosis. The present comparative study reports some innovative features such as FITs’accuracy variation according to T stage across 7 of 8 FITs included at adjusted cutoffs. They also reported for the first time a better accuracy to detect multiple AN compared to T1 CRC across all included FITs.

Few minor revisions could improve the present report’s quality:

  • Was all colonoscopy histopathological examination performed by the same team (gastroenterologist and histopathologist)? IF it is not the case it would have been appropriate to mention if a standardized reading grid was used to adjust for observational biases.
  • Authors should discuss in the “discussion section” all exclusion criteria.
  • In the “method section” authors should precise the method they used for the random selection of non-CRC nor AN’s patients.
  • In my opinion it would have been appropriate to exclude “Eurolyser FOB test”, “OC sensor” and “quick read GO” to minimize measurement’s biases since the first two were not performed in 29 CRC cases and the third aforementioned one was not included in the adjusted cutoffs’ analysis. It would have also been appropriate to mention the 29 CRC cases in the flow chart.
  • Authors should explain why they excluded participant with multiple AN when calculation sensitivities since it could be responsible of selection biases.
  • In my opinion, it would have been appropriate to report clinical and histological characteristics of 29 aforementioned CRC cases and multiple AN’s patients that were excluded
  • In the “result section” Did the authors adjust T stage’s impact on FITs’ accuracy to tumor location. It would be interesting to mention tumor location (distal proximal or rectal) in different T stage Group.
  • In the result section statistical data regarding comparison between T1 cancer and multiple AN’s FITs accuracy are lacking (particularly the p value of the univariate analysis)
  • In the “results” and “discussion” sections, some results that are not statistically significant are fully reported without p-values and make the reading tedious. It would have been appropriate to report them in a concise manner. In my opinion it would have been appropriate to state that no statistical differences were observed regarding UICC stage for 7 out 8 FITS, tumor location, etc.
  • In my opinion it would have been interesting to perform a multivariate analysis to detect independent variables associated with FIT’s accuracy
  • Authors reported results about an ancillary analysis regarding FITs’accuracy according to AN’s size without reporting data in the figures nor providing p values. Can the authors provide supplemental data regarding this concern?

Author Response

Dear Reviewer 1,

thanks alot for your constructive review. Please find attached our point-by-point response. Unfortunately, we needed more time than tought, because our key medical data manger was incapable to work for a long-time period to extract additional data from the colonoscopy/histology reports to conduct the analysis by AA size (Supplementary Table 2).

Kind regards,

Anton Gies

Reviewer 2 Report

The value of FITs related to AA size is reasonable, it should be related to the CRC tumor size (or surface area) also. Why do the authors compare the value of FITs with T statue instead of tumor size (or surface area) ?

Author Response

Dear Reviewer 2,

thanks alot for your time and revision of our manuscript. Please find below this message our reply to your comment.

Kind regards,

Anton Gies

Reviewer 1: The value of FITs related to AA size is reasonable, it should be related to the CRC tumor size (or surface area) also. Why do the authors compare the value of FITs with T statue instead of tumor size (or surface area) ?

Response: We agree that analysis by tumor size would be very interesting. However, our dataset is lacking the size of the tumor in most of the cases